# Probing the ionic defect landscape in halide perovskite solar cells

Sebastian Reichert [1], Qingzhi An[2,3], Young-Won Woo[4,5], Aron Walsh [4,5], Yana Vaynzof [2,3] & Carsten Deibel [1✉]

Point defects in metal halide perovskites play a critical role in determining their properties and optoelectronic performance; however, many open questions remain unanswered. In this work, we apply impedance spectroscopy and deep-level transient spectroscopy to characterize the ionic defect landscape in methylammonium lead triiodide (MAPbI$_3$) perovskites in which defects were purposely introduced by fractionally changing the precursor stoichiometry. Our results highlight the profound influence of defects on the electronic landscape, exemplified by their impact on the device built-in potential, and consequently, the open-circuit voltage. Even low ion densities can have an impact on the electronic landscape when both cations and anions are considered as mobile. Moreover, we find that all measured ionic defects fulfil the Meyer–Neldel rule with a characteristic energy connected to the underlying ion hopping process. These findings support a general categorization of defects in halide perovskite compounds.

[1] Institut für Physik, Technische Universität Chemnitz, 09126 Chemnitz, Germany. [2] Kirchhoff-Institut für Physik and Centre for Advanced Materials, Ruprecht-Karls-Universität Heidelberg, Im Neuenheimer Feld 227, 69120 Heidelberg, Germany. [3] Integrated Centre for Applied Physics and Photonic Materials and Centre for Advancing Electronics Dresden (cfaed), Technical University of Dresden, Nöthnitzer Straße 61, 01187 Dresden, Germany. [4] Department of Materials, Imperial College London, Exhibition Road, London SW7 2AZ, UK. [5] Department of Materials Science and Engineering, Yonsei University, Seoul 03722, Korea. ✉email: deibel@physik.tu-chemnitz.de

Triggered by the first demonstration of a perovskite solar cell in 2009[1], significant research efforts have been devoted to the field of perovskite photovoltaics leading to a record power conversion efficiency of 25.5%[2]. This remarkable performance is made possible by a combination of advantageous properties of perovskite materials, among which most noteworthy are their low exciton binding energies, high absorption coefficients, high charge carrier diffusion lengths and correspondingly long lifetimes of free charge carriers[3–6]. Additionally, significant progress has been made over the last decade in the development of novel fabrication methods and device architectures as well as optimization by interfacial engineering[7–11].

Despite these advancements, several aspects of perovskite solar cells remain a challenge. For example, in many different fabrication approaches, mobile ions have proven to be a major limitation[12–16]. Mobile ions or ionic defects were shown to be the source of current density-voltage hysteresis and were linked to a reduced stability of devices[12,14,17–20]. Moreover, ionic defects that form states within the bandgap which act as recombination centers, can reduce the photovoltaic performance of the device[21,22]. Despite their importance, characterization of ionic defects and their properties in perovskite materials is incomplete. According to calculations and experimental reports, the most likely native point defects in methylammonium lead triiodide (MAPbI$_3$) perovskites are charged vacancies, such as $V_I^+$ and $V_{MA}^-$ and interstitials such as $I_i^-$ and MA$_i^+$[23–26]. Experimentally, ionic defects and their migration have been observed by a range of methods[27–29].

Noteworthy is the work by Futscher et al.[30], who employed transient capacitance measurements on MAPbI$_3$ solar cells to reveal both a fast ($t <$ ms) and relatively slow ($t \sim$ s) species, which varied by several orders of magnitude in both their concentration and diffusion coefficient. While in all their measurements the authors assigned the fast species to $I_i^-$ and the slow species to MA$_i^+$, they also observed variations in activation energies, diffusion coefficients, and ion concentrations when measuring different samples fabricated either in their laboratory or that of others. This observation is not uncommon, especially in light of the wide range of reported defect parameters presented in literature for the same perovskite material[23,30–38]. One contributing factor to this observation is related to the method of evaluation of the transient ion-drift measurements. Recently, we developed an extended regularization algorithm for inverse Laplace transform for deep-level transient spectroscopy (DLTS) that reveals distributions of migration rates for ionic species instead of single migration rates[39]. This finding suggests that in part, the differences and inconsistencies reported in literature can originate from the fact that various experimental methods may probe different parts of the same ionic defect distribution.

Another significant contributing factor, is the high sensitivity of perovskite materials to their fabrication conditions. Subtle changes in the atmospheric environment[40], annealing process[41], or perovskite precursor stoichiometry[42,43] have all been shown to affect the properties of the perovskite layers. These changes will also influence the properties of the ionic defects. For example, the model reported by Meggiolaro et al.[21] describes the dependence of defect formation energies on the microstructure of the perovskite layer and is in good agreement with the experimental results of Xing et al.[44]. Taken together, these observations highlight the need to investigate more deeply the ionic defect landscape in perovskite materials, and identify fundamental processes that govern their formation and physical properties.

In this work, we purposefully tune the ionic defect landscape of MAPbI$_3$ perovskite samples by fractionally modifying the stoichiometry of the perovskite precursor solution. This results in a gradual change in the densities of the various types of defects as suggested by both X-ray photoemission spectroscopy[42] and photoluminescence microscopy measurements[45]. Herein, we directly probe the variations to the ionic defect landscape by impedance spectroscopy (IS) and DLTS, and reveal the interplay between this defect landscape and the electronic landscape of the device. We found, that even small ion densities can have an impact on the electronic landscape. By comparing the ionic migration rates with literature values, we discovered that the systematic variation in our study allows to categorize the results from literature, leading to a remarkably good agreement. Moreover, we show that the temperature dependent diffusion parameters of all the ionic defects fulfill the Meyer–Neldel rule, which we link to the fundamental hopping process of mobile ion transport in halide perovskite solar cells.

## Results

To controllably tune the defect landscape in MAPbI$_3$ perovskite solar cells, we exploited the method developed by Fassl et al[42]. to fabricate a series of samples from precursor solutions with gradually changing stoichiometry. In short, we start by intentionally preparing an understoichiometric solution, in which a slight deficiency of methylammonium iodide (MAI) is expected to result in films rich in vacancies such as $V_I^+$ and $V_{MA}^-$. By gradually increasing the MAI content in the solution, a stoichiometric ratio is reached, followed by a transition to an overstoichimetric regime, in which access of MAI increases the densities of $I_i^-$ and MA$_i^+$ interstitials. Chemical analysis for verification of the composition change with stoichiometry were performed by Fassl et al[42]. on a series of identical samples. To eliminate the influence of different extraction layers, all devices share a common architecture, in which poly(3,4-ethylenedioxythiophene)-poly(styrenesulfonate) (PEDOT:PSS) is used for hole extraction, while [6,6]-phenyl-C$_{61}$-butyric acid methyl ester (PC$_{61}$BM) is used for electron extraction. A thin layer of bathocuproine (BCP) is introduced between the PC$_{61}$BM layer and the Ag contact in order to achieve efficient hole-blocking[46,47]. The current density–voltage (JV) characteristics of the resulting photovoltaic devices are shown in Supplementary Fig. 1. A very small hysteresis, often associated with the presence of mobile ions[15,16,48–50], appears when sweeping in both voltage directions. The solar cell parameters averaged over both scan directions are shown in Supplementary Fig. 2 and are in agreement with the previous report by Fassl et al.[42]. In short, while the fill factor (FF) and short-circuit current ($J_{sc}$) are only very slightly influenced by the changes in stoichiometry, the open-circuit voltage ($V_{oc}$) and consequently the power conversion efficiency (PCE) strongly increase for increasing stoichiometric ratios. For more information about detailed SEM, XRD, and film morphology analysis see ref. [42].

**Capacitance–voltage (CV) profiling.** CV measurements may offer first insights into the ionic defect landscape of the devices. We performed these measurements at an ac frequency of 80 kHz using a fast sweep rate of 30 V/s in the reverse scan direction. Following the methodology of Fischer et al[51]. the devices were pre-biased for 60 s at 1 V, in order to minimize the influence of mobile ions present at the interfaces of the active layer. The results of the CV measurements (Supplementary Fig. 3a) were evaluated using the Mott–Schottky approach[52,53]:

$$1/C^2 = \frac{2(V_{bi} - V)}{e\epsilon_0\epsilon_R N_{eff}}, \qquad (1)$$

where $V$ is the applied external voltage, $e$ is the elementary charge, $\epsilon_0$ is the absolute permittivity and $\epsilon_R$ is the relative permittivity. The relative permittivity can be obtained at reverse bias in the CV

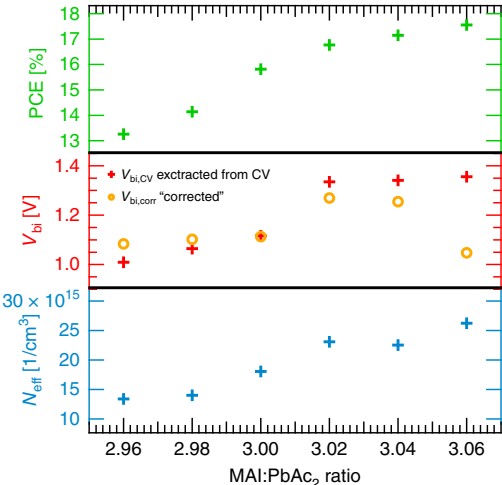

**Fig. 1 Capacitance–voltage parameters in comparison to PCE.** The dependence of power conversion efficiency PCE (green), built-in potential $V_{bi}$ (red) and effective doping density $N_{eff}$ (blue) on stoichiometry (MAI:PbAc$_2$) of the precursor solution is shown. PCE values are taken from Supplementary Fig. 2 for comparison. $V_{bi}$ can be corrected (yellow circles) by estimating the potential drop caused by mobile ions at the interfaces.

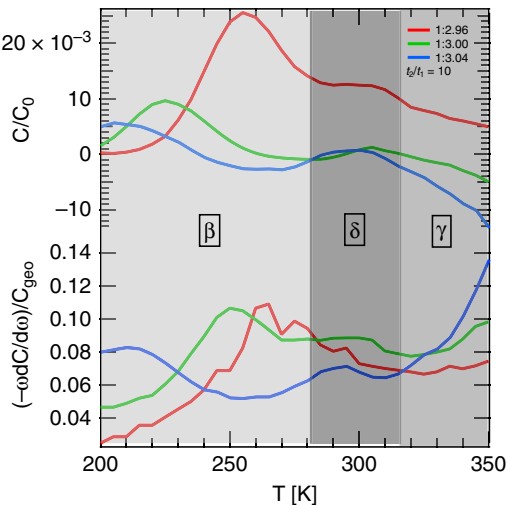

**Fig. 2 Comparison of DLTS and IS spectra.** DLTS spectra (top) and IS spectra (bottom) for the samples with precursor stoichiometry: 2.96 (red), 3.00 (green) and 3.04 (blue) are shown. Areas are shaded differently in order to highlight where each defect ($\beta$, $\gamma$, and $\delta$) is observable. Defect $\gamma$ is not completely visible for the chosen rate window $t_2/t_1 = 10$. For a more detailed overview see Supplementary Figs. 5 and 8.

measurements, where complete depletion can be assumed. In this case the capacitance in this region corresponds to the geometrical capacitance $C_{geo}$, which can be used to calculate the relative permittivity as follows:

$$C_{geo} = \frac{\epsilon_0 \epsilon_R}{d}. \qquad (2)$$

The obtained values are shown in Supplementary Fig. 3b. In the case of high frequencies, the temperature dependence of the capacitance is small and can be neglected. By applying Eq. (1) to the range dominated by the depletion capacitance, the built-in potential ($V_{bi}$) and effective doping density ($N_{eff}$) can be extracted. In Fig. 1, these values are compared to the PCE values from Supplementary Fig. 2. The results indicate that both $V_{bi}$ and $N_{eff}$

increase with increasing stoichiometry. The increase in $V_{bi}$ is in agreement with the findings of Fassl et al.[42], where a shift in the exponential diode characteristics revealed a similar trend in $V_{bi}$. According to literature[54–56], the increase in $N_{eff}$ suggests an overall higher defect density for overstoichiometric samples since ions introduce additional charges and affect the net doping concentration. This is in agreement with the experimental observation of a lower photoluminescence quantum efficiency for samples with higher stoichiometry[45].

**Determining the defect landscape by impedance spectroscopy (IS) and DLTS.** Advanced spectroscopic techniques such as IS and DLTS offer further insights into the defect landscape of the devices. In an IS experiment, the current response to an externally applied alternating voltage at a certain frequency $\omega$ is measured and considered as a capacitance signal by taking into account the imaginary part of the impedance $Z$[57,58]:

$$C = \frac{\text{Im}(1/Z)}{\omega}, \qquad (3)$$

by modeling the solar cell as a capacitor in parallel to a shunt resistance. To obtain a complete picture of the defects and to quantify their physical properties, we performed IS measurements over a wide frequency range (0.6 Hz < $\omega$ < 3.2 MHz) and at different temperatures (200–350 K in 5 K increments).

There are two responses in the representative IS spectra as shown in Figs. 2 and S4: a low frequency response (<10$^2$ Hz) at high temperatures (>315 K) and a step at higher frequencies (>10$^2$ Hz) and lower temperatures (<285 K) for each of the investigated samples. These responses can be assigned to two different defects. Particularly noteworthy is the increase of the low frequency section of the spectra with increasing stoichiometry, which indicates its impact on the properties of the corresponding ionic defect. From the capacitance spectra, we are able to extract the ion (defect) diffusion coefficient $D$ based on the equation:

$$D = D_0 \exp\left(-\frac{E_A}{k_B T}\right), \qquad (4)$$

where the $k_B$ is the Boltzmann constant, $T$ is the temperature, $D_0$ is the diffusion coefficient at infinite temperature and $E_A$ is the activation energy for ion migration. This is done by extracting the ion migration rates (emission rates is the corresponding term from DLTS when applied to study electronic defects in semiconductors)[30,33,59,60] as defined by:

$$e_t = \frac{e^2 N_{eff} D_0}{k_B T \epsilon_0 \epsilon_R} \exp\left(-\frac{E_A}{k_B T}\right), \qquad (5)$$

from the maxima of the derivative—$\omega dC/d\omega$, shown in Supplementary Fig. 5. The presence of two maxima in these spectra reveal two distinct ionic defects, $\beta$ and $\gamma$. We summarized the migration rates associated with these two defects in an Arrhenius diagram (Supplementary Fig. 6) and calculated the activation energies $E_A$ and the diffusion coefficients $D_{300 K}$ at 300 K based on Eq. (5).

Interestingly, the defects $\beta$ and $\gamma$ show opposing trends in terms of $E_A$, $D_{300 K}$, and $N_{ion}$ with varying stoichiometric ratios (Fig. 3). For $\beta$, the activation energy decreases whereas the diffusion coefficient at 300 K increases for increasing stoichiometry, whereas $\gamma$ shows the inverse behavior. We conclude that by increasing the sample stoichiometry, ion migration of defect $\gamma$ is suppressed, while the defect $\beta$ becomes more mobile.

The accumulation of mobile ions at the interfaces of the active layer, driven by the internal electric field of the photovoltaic devices, was reported in several studies[15,50,61,62]. The resultant inhomogeneity of the ionic distribution in the perovskite active

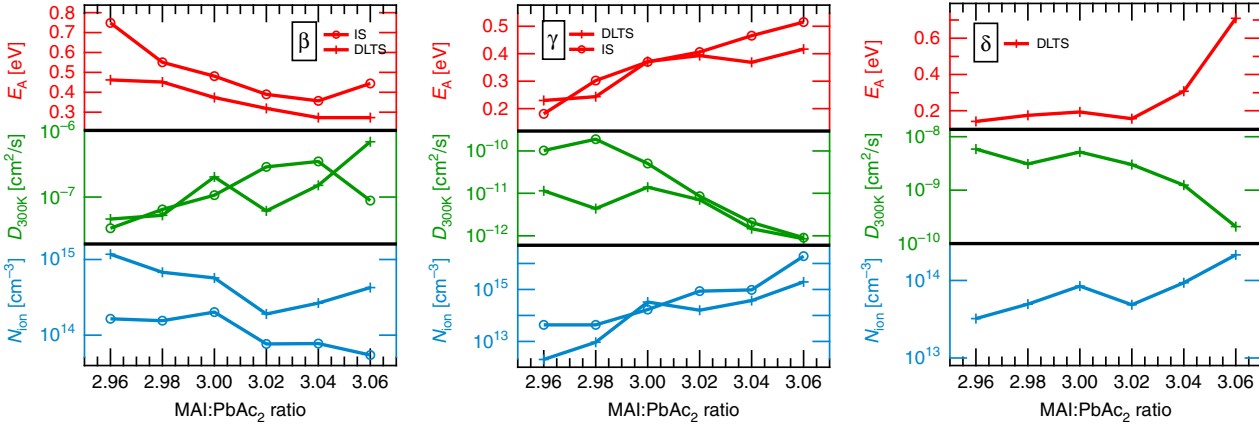

**Fig. 3 Defect parameters for ionic species $\beta$, $\gamma$, and $\delta$.** Activation energy $E_A$ (red), diffusion coefficient $D_{300\,K}$ at 300 K (green) and ionic defect concentration $N_{ion}$ (blue), extracted by IS (circles) and DLTS (crosses) for all perovskite solar cells with stoichiometric ratio of precursor solution (MAI: PbAc$_2$).

layer leads to the formation of a Debye layer of cations at the hole transport layer and a Debye layer of anions at the electron transport layer[63,64]. As a result of the inhomogeneity, the defect density from IS measurements cannot be determined by using the approach by Walter et al[65]. for semiconductor defects. A more feasible approach can be found by taking into account the capacitance of the ionic Debye layer [66]:

$$N_{ion} = \frac{k_B T \Delta C^2}{e^2 \epsilon_0 \epsilon_R}. \tag{6}$$

In this case, $\Delta C$ is proportional to the capacitance step observed in Supplementary Fig. 4. Following this approach, the ionic defect concentration of $\beta$ slightly decreases with increasing stoichiometry, while $N_{ion}$ of $\gamma$ shows a notable increase as shown in Fig. 3. We conclude that defect $\gamma$ dominates the behavior of overstoichiometric samples, while the more mobile defect $\beta$ dominates the understoichiometric ones.

To expand the insights gained by IS, we performed DLTS measurements on the same set of solar cells. For DLTS, a voltage filling pulse (from 0 to 1 V for a duration of 100 ms) is applied to the devices, while measuring the capacitance response at 80 kHz until the solar cell returns to equilibrium conditions[58,67]. During the filling pulse, mobile ions are pushed from both interfaces of the perovskite layer into the perovskite bulk until they reach a new steady state condition. After the filling pulse, the mobile ions move back to the interfaces caused by the internal field, which introduces a change of the solar cell capacitance[30]. The resulting transients, shown in Supplementary Fig. 7, were averaged over 35 single measurements to yield a high signal-to-noise ratio, and were measured within the same temperature range as the IS measurements. For the evaluation, we performed the commonly utilized boxcar method[67] as shown in Figs. 2 and S8.

The analysis of the DLTS data reveals three different temperature-dependent peaks associated with three distinct defect states. Two of these defects exhibit high migration rates at low to medium temperature range, while the third shows low migration rates at higher temperatures. Following the good agreement in the peak position shown in Fig. 2 and that of the migration rates plotted in the Arrhenius diagram (Supplementary Fig. 6), we conclude that one of the two defect states with high migration rates corresponds to defect $\beta$ previously identified by IS. The defect exhibiting low migration rates is attributed to $\gamma$ in agreement with IS. Similarly to the IS data, defect $\gamma$ dominates the boxcar spectrum for high stoichiometry samples. The remaining defect with comparably high migration rates was labeled $\delta$. This

defect cannot be evaluated with IS as it is only observable as a shoulder, not a peak, and only for higher stoichiometric ratios in a very narrow temperature range.

Unlike IS, DLTS data allows to distinguish between positive and negative ionic defects, i.e., anions and cations. As shown in Fig. 2 and S8, defects $\beta$ and $\delta$ have a positive sign and correspond therefore to anions, whereas $\gamma$ corresponds to a cation. As mentioned above, the migration rates ($e_t = 1/\tau$) of these ionic defects can be extracted from the position of the peaks shown in Supplementary Fig. 8 and complement the results of IS measurements when plotted in the same Arrhenius diagram (Supplementary Fig. 6). We note that since the slow response of $\gamma$ dominates for overstoichiometric devices, the transients for these devices at very high temperatures did not return to equilibrium within the recorded transient time length of 30 s (Supplementary Fig. 7). We excluded these non-equilibrium transients from the determination of defect parameters, as they lead to overestimated migration rates for a given temperature.

The overall trend of $E_A$ and $D_{300\,K}$ for defects $\beta$ and $\gamma$, shown in Fig. 3, is comparable with the results obtained by IS. We note that while defect parameters extracted using IS and DLTS exhibit the same general trend, they do show some variance in the absolute values of the extracted defect parameters. These differences, which are visible as an offset in the defect parameters, might arise from the broad distributions of ionic defects, reported in our recent work[39]. Different parts of the same defect distribution are probed by each of the two methods. Furthermore, the difference between IS and DLTS can be caused by the fact that the mobile ions with IS are detected when they reside near the interfaces at 0 V DC bias, whereas with DLTS, the mobile ions are probed while they move from the bulk to the interface. For defect $\delta$, identified solely via DLTS, we observe a significant increase in $E_A$ for overstoichiometric samples, accompanied by a strong decrease in $D_{300\,K}$. The ionic defect concentration, $N_{ion}$, can be extracted from DLTS measurements by using the ratio between the capacitance change $\Delta C$ caused by the ionic movement and the steady state capacitance $C_\infty$, given by:

$$N_{ion} \propto \frac{\Delta C}{C_\infty} N_{eff} \text{ if } N_{ion} \ll N_{eff}. \tag{7}$$

As shown in Fig. 3, the trend of $N_{ion}$ for defects $\beta$ and $\gamma$ with changing stoichiometry is also in agreement with the results obtained with IS. For defect $\delta$, the ionic defect concentration is found to increase with stoichiometry, similar to the behavior of defect $\gamma$. Due to the fact that the diffusion coefficient of $\gamma$ is relatively small, it cannot be ensured that this ionic species

reaches equilibrium within the duration of the filling pulse. As a result, the determined ion density for $\gamma$ represents only a lower limit.

Since we performed reverse-DLTS measurements in our recent work to distinguish between electronic and ionic defects[39], we attribute all observed defects to mobile ions. As part of our scenario in this recent work, we assign the anion $\beta$ to $V_{MA}^-$ and $\delta$ to $I_i^-$. The cation $\gamma$ is attributed to $MA_i^+$. This assignment is in good agreement with the results of Fassl et al.[42], where XPS measurements showed an increase in the I/Pb and N/Pb ratios with increasing stoichiometry. We note that the assignment of cation $\gamma$ to $MA_i^+$ may appear in contrast to the reports by Maier and coworkers that claim that methylammonium cations are only mobile in terms of reorientation, ruling out the migration of this species[68,69]. However, it was shown that rotational dynamics of methylammonium cations occurs with relaxation times in the ps timescale at room temperature[70–72], which would be too fast to explain hysteresis. Other groups propose that methylammonium can slowly migrate[15,31,54], since other possible cations, such as iodine vacancies, are expected to have far higher diffusion coefficients[32,73,74]. Nevertheless, we stress that DLTS provides information solely on the charge of the ionic defects and cannot directly determine the specific ionic species. Only electrically charged species can be observed. We therefore exclude neutral protonic species of MAI as reported in literature[75] to be the origin of the observed mobile ions.

**Interplay between the ionic and electronic landscapes**. The mixed ionic–electronic conducting nature of perovskites dictates that the ionic and electronic landscapes of these materials cannot easily be decoupled[76,77]. One aspect linking the two is related to the effect of ion accumulation at the interfaces of the perovskite layer and the extraction layers that sandwich it[78]. Such ionically charged interfacial layers influence the internal electric field and the built-in potential of the device, suggesting that the estimation of $V_{bi}$ from CV measurements as discussed above needs to be re-evaluated[79]. The validity of the Mott–Schottky relation (Eq. 1) is based on the assumption that the charge carrier density within the perovskite layer is homogeneously distributed, which may not be the case for perovskite solar cells. While we prebiased the devices before measuring CV in an attempt to eliminate the accumulation of ions at the interfaces, the resultant trend in $V_{bi}$ is consistent with what has been observed by diode J–V characterization, for which no pre-biasing was applied[42]. This might indicate that ions still accumulate at the interfaces, resulting in a voltage drop that changes the $V_{bi}$. A simple model that accounts for this voltage drop can be constructed by considering these interfacial ion densities as Debye layers[63,66,78]. The overall charge for one ionic species can be expressed by $\Delta Q = eN_{ion}L_D$, where $L_D$ is the Debye length according to:

$$L_D = \sqrt{\frac{\epsilon_R \epsilon_0 k_B T}{e^2 N_{ion}}}. \quad (8)$$

In order to account for the potential drop $\Delta V$ caused by the mobile ion density of cations $N_C$ and anions $N_A$, we assumed a series connection of the capacitance caused by cations $C_C$ and anions $C_A$,

$$\Delta V = \sqrt{\epsilon_R \epsilon_0 k_B T \left(\frac{1}{C_C} + \frac{1}{C_A}\right)} (\sqrt{N_C} - \sqrt{N_A}). \quad (9)$$

With Eq. (9) the corrected $V_{bi,corr}$ can be obtained by correcting the determined built-in potential by CV measurements with the voltage drop caused by mobile ions at the interfaces:

$$V_{bi,corr} = V_{bi,CV} - \Delta V. \quad (10)$$

The capacitance responses $C_A$ and $C_C$ by anions and cations, respectively, can be estimated by the capacitance steps in the IS spectra from Supplementary Fig. 4, as they correspond to the ion density $N_{ion}$ according to Eq. (6). Taking into consideration the ionic interfacial layers to suppress the trend observed in $V_{bi}$, a more consistent value of around 1.1 V can be obtained as shown in Fig. 1. This result is more in line with our expectations, since all the devices share the same extraction layers and contacts. As shown in a recent study[77], a shift in $V_{bi}$ can also be caused by electronic charge carrier accumulation at the hole transport layer interface. While our calculation correct the influence of ions on $V_{bi}$, we cannot rule out an additional electronic influence. However, the result of our correction suggests that, here, the consideration of ions is sufficient. We note that we use a model assuming two mobile ionic species to correct the shift of the built-in potential in contrast to several studies, where only one mobile species is assumed[64,66]. Accordingly, the magnitude of the defect density necessary to cause band bending reported here is lower compared to these studies (for more information, see SI).

One interesting, and seemingly contradicting, observation is related to the observed increase in $V_{oc}$, which coincides with an increase in the overall ionic defect density with increasing stoichiometry. Recent studies suggest that mobile ions may act as non-radiative recombination centers[21,22,80], evidenced, for example, by a decrease in the photoluminescence quantum efficiency (PLQE). Indeed, overstoichiometric samples exhibit a markedly lower PLQE than understoichiometric ones[45]. Based on these results, one might expect for overstoichiometric devices lower open-circuit voltages than for understoichiometric ones[81], in contrast to the experimental observation shown in Supplementary Fig. 2. However, this apparent discrepancy can be reconciled when taking into account the substantial increase in $V_{bi}$ with higher stoichiometric ratio. Consequently, while a high ionic defect concentration has a negative effect on $V_{oc}$ due to increase of non-radiative recombination, this effect is weaker than the considerable increase introduced by changes to the energetic alignment between $MAPbI_3$ and the transport layers and the resultant change in $V_{bi}$[42]. Moreover, the impact of ions on the energy landscape is in agreement with a recent study by the group of Maier[82], where the authors report an increase of band bending in $MAPbI_3$ toward the electron transport layer originating from an ionically dominated space charge. The interplay of ions and the space charge potential enable device improvements by interfacial engineering.

The intricacy of the interplay between the ionic and electronic landscapes is exemplified by plotting the $E_A$ versus the $V_{bi,CV}$ as shown in Fig. 4. The $E_A$ of defects $\beta$ and $\gamma$ are of similar magnitude and show a broadly linear dependence on the built-in potential, albeit with slopes of opposing signs. Straight dashed lines with slopes of $\pm 1$ were added to Fig. 4 as a guide to the eye. These similar, but opposing trends in slope supports our earlier assignment of these defects to be related to the same type of MA ion. We can associate defect $\beta$ with an MA vacancy with negative charge, and $\gamma$ with a positively charged MA interstitial. Our interpretation of defect $\delta$ cannot be confirmed in this manner, since we do not observe the corresponding defect species with an opposing charge.

The dependence of $E_A$ on $V_{bi}$ might be a consequence of the band bending introduced by the interfacial ion accumulation. As the ion concentration increases, stronger band bending at the interfaces leads to higher fields that impedes the ionic hopping process at the interfaces lowering their overall mobility. This explanation is supported by plotting $E_A$ and $D_{300 K}$ versus $N_{ion}$ (see Supplementary Fig. 9). Although the trends are less clear, we generally observe an increase of $E_A$ and a decrease in $D_{300 K}$ for higher defect concentrations. The dependence is in agreement

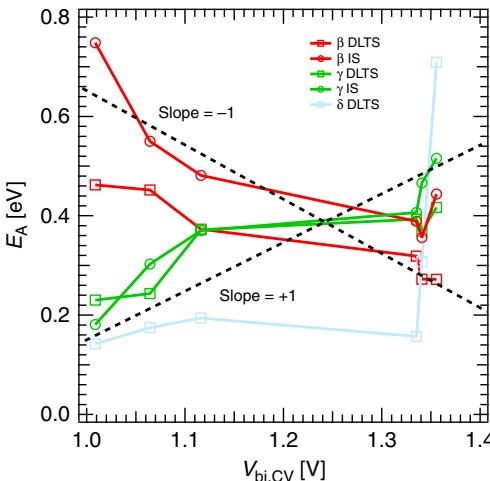

**Fig. 4 Activation energy versus built-in potential.** Activation energy $E_A$ of the ionic defects $\beta$ (red), $\gamma$ (green), and $\delta$ (blue) plotted in dependence of the built-in potential $V_{bi,CV}$ extracted from CV measurements for all investigated perovskite solar cells. The dashed lines with slopes of ±1 are guides for the eye.

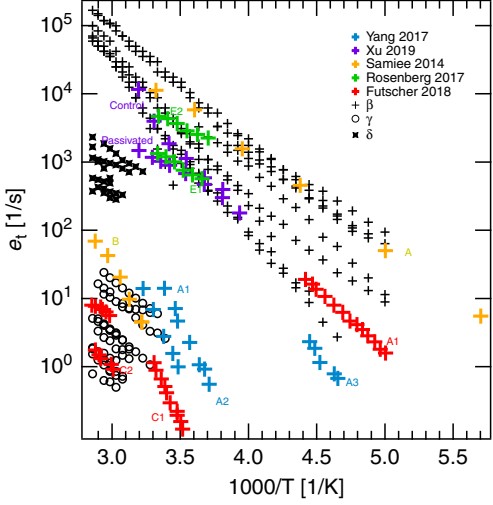

**Fig. 5 Literature comparison.** Migration rates reported in literature[30,34–36,38] (often called emission rates in the papers) were plotted in an Arrhenius diagram for comparison to our findings (species $\beta$ (crosses), $\gamma$ (circles), $\delta$ (stars)). All reported emission rates can be associated with two regimes at low emission rates and high temperatures and at high emission rates at high and middle temperatures.

with our finding that mobile ions are pushed stronger towards the interfaces caused by the relation between the internal electric field and the density of ions. As a result, the diffusion coefficient decreases with higher ion density. This interaction between the ionic and electronic landscapes highlights the need to construct a clearer picture of the underlying defect physics in perovskite devices. We point out that the observed trends in activation energy, diffusion coefficient, and defect density cannot be explained by morphology changes, as measurements on an identically made set of samples show only a slight change of the grain size by a few percent, which we exclude to be the origin of the change in the defect parameters by several orders of magnitude[42].

**Unraveling the defect landscape across the literature.** To evaluate our results in a broader context, we compared the migration rates measured herein, with data available from literature. We

chose several studies with similar measurement methods such as DLTS and IS, but with a selection of different perovskite materials and transport layers as summarized in Table S10. Included in Fig. 5 are the results of Samiee et al.[36], who observed two different defects in a mixed halide perovskite using IS, and three defects (attributed to cations) probed by Yang et al.[34] using DLTS on FAPbI₃. Additionally, included are the emission rates of two defects measured using current DLTS by Rosenberg et al.[35] in MAPbBr₃ single crystals and those probed by Xu et al.[38] on FAPbI₃ light-emitting diodes. Finally, the results of Futscher et al.[30] using transient ion-drift measurements (which is DLTS under a different name) were added, which exhibit two ionic species assigned as $I_i^-$ and $MA_i^+$ interstitials in MAPbI3 solar cells.

This comparison reveals a remarkable agreement between reports despite the use of different perovskite compositions and device structures. The reported emission rates broadly fall into two categories: those with low emission rates at high temperatures or those with high emission rates at high or medium temperatures. This assessment indicates that there are most likely two dominant underlying ionic defects, which can be universally observed in all perovskite materials investigated thus far.

**Meyer–Neldel rule.** To gain an understanding of the underlying mechanism for ion transport, we examine the relationship between the diffusion coefficient $D_0$ (at infinite temperature) and the activation energy $E_A$ according to Eq. (4) and Fig 3. Figure 6a reveals a clear linear dependence between these two values for each of the ionic defects. Such a linear relation is known as the Meyer–Neldel rule, which is often used to describe thermally activated processes[83]. According to the Arrhenius Eq. (4), the Meyer–Neldel rule states that the prefactor $D_0$ itself depends on the activation energy $E_A$ via:

$$D_0 = D_{00} \exp\left(\frac{E_A}{E_{MN}}\right) \text{ with } E_{MN} = k_B T_{MN}, \qquad (11)$$

where $D_{00}$ refers to the critical diffusion coefficient, $E_{MN}$ is the characteristic energy and $T_{MN}$ is the corresponding characteristic temperature. Equation (11) yields very similar values for $E_{MN}$: 28 meV, 30 meV, and 35 meV for $\beta$, $\delta$ and $\gamma$, respectively. The critical diffusion coefficient $D_{00}$ of $\beta$ with $3 \times 10^{-7}$ cm²/s is one order of magnitude higher than for $\delta$ ($1 \times 10^{-8}$ cm²/s). $\gamma$ has the lowest $D_{00}$ which is equal to $4 \times 10^{-10}$ cm²/s. As a consequence of the Meyer–Neldel rule, the migration rates shown in Supplementary Fig. 6 and the diffusion coefficients (presented in Fig. 6b) that are associated with the same defect, intersect at $1000/T_{MN}$. At this intersection point, which is different for each defect species, the migration rates become independent of stoichiometry. In other words, the ionic defect landscape is no longer affected by stoichiometry at $T_{MN}$. As guide to the eye, we added dashed lines to Fig. 6b, which were extracted from the values of the fits presented in Fig. 6a.

We propose two possible origins for the Meyer–Neldel behavior. The first is based on disordered organic semiconductors, where the legitimacy of the Meyer–Neldel rule was linked to a Gaussian distribution of defect levels or hopping energies[84]. Despite being crystalline materials, it is well established that halide perovskites contain a significant amount of disorder due to spatial and temporal variations of octahedral tilts and molecular rotations[85]. A range of defect environments and transition pathways are therefore expected. Indeed, in our recent work, we demonstrated a distribution of migration rates for each of the reported defects[39]. This is also supported by a combined experimental–theoretical work, where modeling of ion migration induced PL quenching was only possible by applying a Gaussian distribution of ion migration rates[86].

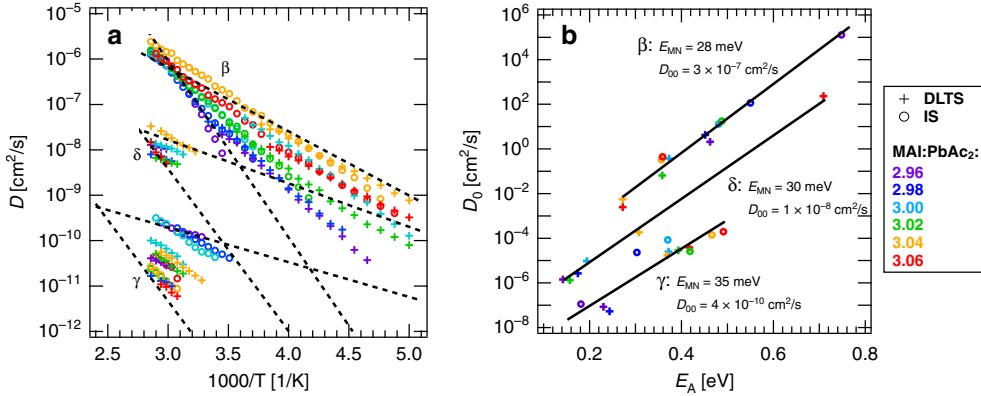

**Fig. 6 Meyer–Neldel rule in two representations. a** Arrhenius plot with diffusion coefficients over the temperature. Additional dashed lines, which correspond to the start and end points of the fits (solid lines) from the Meyer–Neldel plot (**b**), illustrate the range and common nature of diffusion coefficients for each ionic defect ($\beta$, $\gamma$, and $\delta$) and for all precursor stoichiometric ratios. **b** Meyer–Neldel plot: diffusion coefficients at infinite temperature $D_0$ plotted over the corresponding activation energy $E_A$. The characteristic energy $E_{MN}$ and the critical diffusion coefficient $D_{00}$ can be extracted from the linear fits (solid lines).

A second explanation arises if a multiexcitation entropy model is considered. A single hopping event is usually the result of a multiphonon excitation, since the activation energy for ion migration is large compared to the phonon energy (e.g., 16.5 meV for optical phonons)[87,88]. Consequently, a large number of activation pathways are available for each hopping event. A higher activation energy results in a larger number of distinct pathways expressed by the entropy, which is proportional to the exponential prefactor $D_0$[89,90]. Based on this model, the Meyer–Neldel rule originates from the absorption of $N_A$ phonons with $N_A \cdot E_{ph} = E_A$, which transfers the ion first to an activated state, and then—accompanied by the emission of multiple phonons—to the target site of the perovskite lattice[91].

To probe the atomistic nature of a typical diffusion process, we performed first-principles calculations of charged vacancy migration in the room temperature phase of MAPbI$_3$ using the technical setup reported elsewhere[31]. We consider a low energy transition in the (001) plane as illustrated in Fig. 7. The associated migration barrier is 0.55 eV and it follows a curved diffusion pathway. Even in a single plane, due to the presence of MA, the initial and final states differ in energy by 60 meV, which supports the first disorder explanation. We further determine the vibrational frequency at $T = 300$ K around using `CarrierCapture`[92]. Effective frequencies of 0.4–0.7 THz represent the curvature of the potential energy surface along the directions of ion diffusion. These are unusually soft owing to a combination of the heavy elements and the flexible perovskite structure. A simple estimation of $N_A$ suggests that hundreds of phonon modes are involved in a single hopping process, which supports the second explanation. While we cannot yet assign the validity of the Meyer–Neldel rule to a single origin, it offers interesting insights into the physical mechanisms of ion migration in halide perovskites.

## Discussion

In summary, we investigated the ionic defect landscape of MAPbI$_3$ samples with gradually varying defect densities, introduced by fractionally varying the stoichiometry of the perovskite precursor solution (MAI:PbAc$_2$). By combining the results of IS and DLTS measurements, we identify three ionic defect, which we attribute to $V_{MA}^-$, $I_i^-$, and MA$_i^+$. We explore the tight link between the ionic defect and electronic landscapes in perovskite devices, and reveal that the accumulation of defects at the interfaces of the perovskite layer results in an increase of the built-in potential for

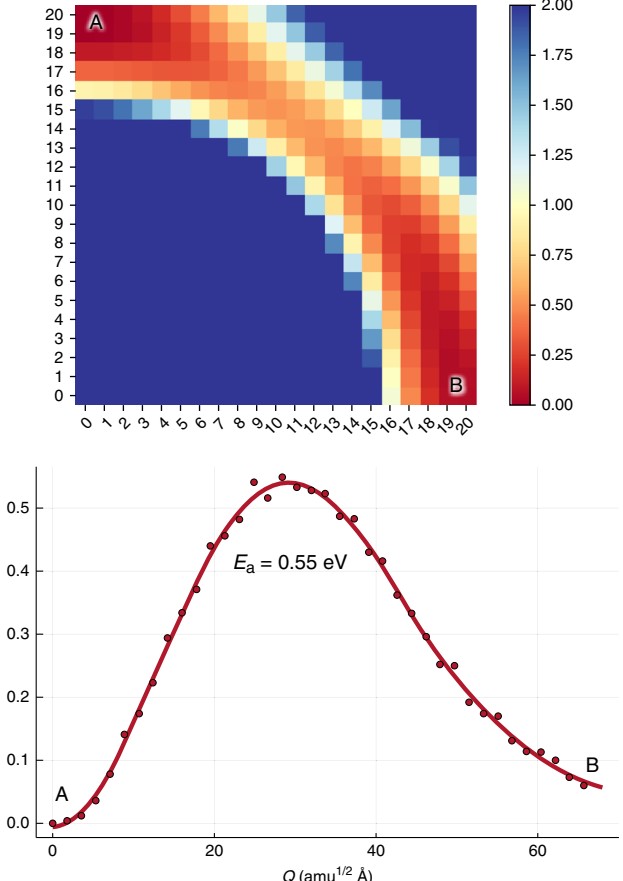

**Fig. 7 First-principles analysis of the activation energy for hopping. a** Calculated 2D energy surface on a 20 × 20 real-space grid for $V_I^+$ migration in a (001) plane of tetragonal MAPbI$_3$ from initial state A to final state B along the configurational coordinate Q. **b** The projected 1D pathway showing the associated activation energy.

increasing stoichiometry, which we show to be the dominant factor influencing the open-circuit voltage of the devices. Interestingly, the interplay between electronic landscape and mobile ions are found to be present at even small ion densities when

considering both type of ions as mobile. The presence of ionic interfacial layers is also shown to affect the $E_A$ of the various defects, by impeding their transport due to the high electric fields they introduce. We compared the temperature dependent ion migration rates to the literature, and were able to categorize defect parameters of different perovskite materials and device architectures. Importantly, we find that the ionic defects we observed fulfill the Meyer–Neldel rule. We propose that the origin of the Meyer–Neldel rule lies either in the distribution of migration pathways or the multiphonon emission process that characterizes the hopping of ions. Our results offer significant insights into the defect physics of perovskite materials and progress the current understanding of the underlying processes, that govern the properties of this phenomenal class of materials.

## Methods

**Device fabrication**. Pre-patterned indium tin oxide (ITO) coated glass substrates (PsiOTech Ltd., 15 Ω/cm²) were ultrasonically cleaned with 2% Hellmanex detergent, deionized water, acetone, and isopropanol, followed by 10 min oxygen plasma treatment. Modified poly(3,4-ethylene-dioxythiophene):poly(styrenesulfonate) (m-PEDOT:PSS) was spin cast on the clean substrates at 4000 rpm for 30 s and annealed at 150 °C for 15 min to act as hole transport layer[93]. To fabricate devices with varying stoichiometry, a MAPbI₃ precursor solution was prepared by adding MAI (greatcell solar materials) and lead acetate dehydrate (Sigma Aldrich) in a molar ratio of 2.96 to anhydrous N,N-dimethylformamide (DMF, Sigma Aldrich) at a concentration of 42 wt%. Hypophosphorous acid (HPA, Sigma Aldrich) was added to the precursor solution (6.47 µl per 1 ml DMF). To increase the stoichiometry, a pure 2.4 M MAI solution (in DMF) was prepared, and based on the target stoichiometry (2.98 to 3.06 in 0.02 steps), different amounts of the pure MAI solution were added to the 2.96 precursor solution. For further details please see ref. [42]. The perovskite solution was spin cast at 2000 rpm for 60 s in a dry air filled glovebox (relative humidity <0.5%). The prepared samples were transferred to a nitrogen filled glove box, where an electron transport layer PC₆₁BM, 20 mg/ml dissolved in chlorobenzene, was dynamically spin cast at 2000 rpm for 30 s on the perovskite layer followed by a 10 min annealing at 100 °C. Sequentially, a bathocuproine (0.5 mg/ml dissolved in isopropanol) hole blocking layer was spin cast on top of the PC₆₁BM. The device was completed with a thermally evaporated 80 nm thick silver layer.

**Current–voltage characterization**. The current density–voltage (JV) characteristics were measured by a computer controlled Keithley 2450 Source Measure Unit under simulated AM 1.5 sunlight with 100 mW/cm² irradiation (Abet Sun 3000 Class AAA solar simulator). The light intensity was calibrated with a Si reference cell (NIST traceable, VLSI) and corrected by measuring the spectral mismatch between the solar spectrum, the spectral response of the perovskite solar cell and the reference cell. All measurements were performed at room temperature (300 K) with a scan rate of 0.25 V/s. To verify that the samples did not degrade during the experiment, JV measurements were performed both before and after the characterization by defect spectroscopy.

**Defect spectroscopy measurements**. All defects were measured using a setup consisting of a Zurich Instruments MFLI lock-in amplifier with MF-IA and MF-MD options, a Keysight Technologies 33600A function generator and a cryo probe station Janis ST500 with a Lakeshore 336 temperature controller. We performed the defect spectroscopy in the temperature range of 200–350 K in 5 K steps, controlled accurately within 0.01 K, using liquid nitrogen for cooling. DLTS, IS and CV measurements were done applying an AC frequency of 80 kHz with amplitude of $V_{ac} = 20$ mV. For DLTS, the perovskite solar cells were biased from 0 to 1 V for 100 ms. The transients were measured over 30 s and averaged over 35 single measurements. For CV profiling, the solar cells were pre-biased at 1 V for 60 s and rapidly swept with 30 V/s in reverse direction. All measurements were performed on two different batches for proving repeatability.

**Reporting summary**. Further information on research design is available in the Nature Research Reporting Summary linked to this article.

## Data availability

The data that support the findings of this study are available in Zenodo with the identifier https://doi.org/10.5281/zenodo.4049791.

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

## Acknowledgements

C.D. and S.R. acknowledge financial support by the Bundesministerium für Bildung und Forschung (BMBF Hyper project, contract no. 03SF0514C) and thank their project partners from the University of Würzburg and ZAE Bayern for interesting discussions. Y.W.W. thanks Sunghyun Kim for assistance. Via our membership of the UK's HEC Materials Chemistry Consortium, which is funded by EPSRC (EP/L000202), this work used the ARCHER UK National Supercomputing Service (http://www.archer.ac.uk). This work was also supported by a National Research Foundation of Korea (NRF) grant funded by the Korean government (MSIT) (no. 2018R1C1B6008728). Y.V. and C.D. thank the DFG for generous support within the framework of SPP 2196 project (PERFECT PVs). This project has received funding from the European Research Council (ERC) under the European Union's Horizon 2020 research and innovation programme (ERC Grant Agreement no. 714067, ENERGYMAPS).

## Author contributions

Y.V. and C.D. conceptualised the study. Q.A. fabricated the photovoltaic devices and characterised their photovoltaic performance under the supervision of Y.V. S.R., and C.D. planned, and S.R. performed, the defect spectroscopy measurements. Y.W.W. and A.W. performed the first-principles modelling and analysis. S.R. wrote the manuscript with input and revisions by C.D. and Y.V. All authors contributed by discussion to the manuscript.

## Funding

## Competing interests

The authors declare no competing interests.
