## [Peer Review File · Nature Communications]

Reviewers' comments:

Reviewer #1 (Remarks to the Author):

In this paper ionic defects in halide perovskite solar cells are investigated using impedance spectroscopy and DLTS. By adjusting the precursor stoichiometry the amount of induced defects is manipulated, which gives an indication regarding the origin of the defects. An interesting finding is that slow and fast ions are found with a related effect on the built-in potential, indicating that they originate from the same type of MA ion. Regarding the interpretation several questions remain: -As shown in Figure S4 the capacitance, and therefore also permittivity is strongly frequency and temperature dependent due to the ion movement. Equations (4) and (5) contain the permittivity, it is not clear which permittivity values are used for the slow and fast(er) ions and how the temperature dependence of the permittivity is taken into account in the activation energy for the ion diffusion (Eq. (4)).

-The amount of ions calculated (EQ. (5), Figure 3) are only in the 10^{14} - 10^{15} cm⁻³ concentration range. This is considerably lower than the amount of charge carriers generated in a solar cell. As a result, effects of ions on band bending and built-in voltage would be overruled by movement of the charge carriers which would compensate these effects. What is missing as a validation of the obtained numbers are numerical calculations of the band bending and electric field profiles.

Reviewer #2 (Remarks to the Author):

The report of Reichert et al. uses impedance and deep level transient spectroscopies to "offer ... insights into the defect physics of perovskite materials...".

Careful measurements are done on devices, rather than clean materials, to deduce information on the active, perovskite material, which is assumed to be clean in and out because of its mode of preparation, which, even though it probably affects the surfaces and interfaces rather than the bulk, does appear to do so controllably!

DLTS and halide lead perovskites are both hard to control, as materials, and as a method, because of the need to prepare the systems in an initial state that is well-controlled and the need for contacts. The ref. cited suggests such a method and notes its results on the FA Pb perovskite, and its problem with the MA one; the onus of showing that here the result that presumably is obtained, a p-n junction, remains stable during all of the measurements, is on the authors.

The main problem with the report is that it fails to distinguish between clear experimental observables and inferred phenomena, something that, admittedly, is hard and is a general issue in much of scientific research, but seems here to be quite significant. A simple example is the caption of Fig. S1, which contains 2 sentences, one about observables and the other a conclusion that gives, though, the impression of being an observable.

The samples are likely polycrystalline films with a certain grain size distribution, possibly with a respectable area to volume ratio. This can make the interfaces a significant part of the film and, while this is certainly known, the conclusions that are drawn give the impression as referring to the defect physics of the bulk material. If that is not the case, that is not clear.

The Meyer-Neldel rule part shows the data that suggest this effect. However, the authors come up against what has been the main problem with this rule, a clear physical explanation that is testable and gives new insight. In the discussion (there is a problem re. ref. 85, which is for inorganic materials) disorder is suggested and multi-phonon excitation. Possible, but maybe it is possible to

first test it against a more intuitive possibility that can be found in papers from 1968, 1970 and 1986, by B Rosenberg, Kemeny & Rosenberg and JC Dyle, respectively? After that fails, then the above-noted two options can be further considered and content be provided for the statement that they "offer interesting insights into the physical mechanisms of ion migration in halide perovskites".

None of this, though, need in the end, prevent publishing this work, after shrinking, including making parts less verbose, and clearly separating experimental observables from the rest.

Remains that it is hard to understand why this should be in Nat. Comm. (also after re-reading the purpose of the journal). Why not submit to an Applied Physics journal, or if the word "applied" is to be shunned, to PRMaterials, for example?

Reviewer #3 (Remarks to the Author):

To authors

This investigation is trying to figure out the behaving defect formation and the ionic migration in MAPbI₃ with the controlling minor concentrations of MAI molecule inducement by EIS and DLTS analysis. The resulting values are explained in Meyer-Neldel rule which is very traditional theory for microscopic explanation of compensation on activation energy increase. I am sure this investigation that has high novelty of understanding defect formation and migration behavior on MAPbI₃. This article is major revision as mentioned below list.

1. Authors do not show the crystal properties by SEM and XRD. In particular, the SEM surface images are remarkably an important for obtained DLTS results. Because of, the trap concentrations at interface between electrode and perovskite layer is also one of key factor to enhance of deep trap level. (Science 20 Mar 2020: Vol. 367, Issue 6484, pp. 1352-1358). The perovskite film morphologies with fractions of MAI with PbAc₂ should be providing them in SI. When the film morphologies are looks very similar or same, we can begin to discuss on DLTS and EIS results. I can expect the significant morphology differences based on PCE distributions with fraction of MAI. Please should be defined film morphologies.
2. Authors have used fraction of MAI 2.96 to 3.06, which is changed very tiny concentration of MAI for formation of MAPbI₃. The controlling MAI concentration as addressed quantities should be leading to change MA and I defect sites in our expectation. However, the formation of MAPbI₃ using precursor is not able to define as our expectation, for example, the various fraction of MAI with a concentration of PbAc₂ compounds in polar solvent is on stoichiometric to non-stoichiometric status. 1 : 3 of PbAc₂ : MAI is stoichiometric precursor but others are not. The stoichiometric of elemental coordination is not related with using fraction between MAI and PbAc₂ compounds. The most important point of perovskite material before this investigation have to be defined chemical analysis first. When you can define the differences of chemical quantification on organic, Pb and I, DLTS results can explain. Please shows quantified elements on the consisting chemicals on MAPbI₃ using XPS or another possible chemical analysis.
3. In previous investigation (J. Am. Chem. Soc. 2020, 142, 13, 6251–6260), the very small concentration of tri-iodide addition as less than 50 mM is appeared to form dense MAPbI₃ film. It is also obtained to enhance PCE and Voc as addressed in author's investigation. I can presume that MAI addition into MAPbI₃ is leading to increase iodide site in the perovskite crystal comparing to defect passivation by MA site.
4. If authors will have obtained full chemical analysis, you will be understanding the residual Ac molecules are remained in MAPbI₃ film. This molecule can be playing role as reduction of anion site defect. But we do not know the remained concentrations of Ac molecules with fraction of MAI. If you can find out this point, I promise this investigation will be excellent for more wide audience in this research area.
5. One more point of view, the localized MA in MAPbI₃ has two kinds of phases as protonic and

neutral molecules. (Appl. Phys. Lett. 108, 073901 (2016)), (J. Am. Chem. Soc. 2017, 139, 46, 16462–16465) The located those MA molecules are being on different defect formations. In particular, two MA molecule can be acting dynamics with different temperatures of environments such as 200K to 350K. The dynamics of MA from order to disorder phases are contributed to change inorganic crystal frames from disorder to order. Namely, DLTS results should be considered to interpret this point of view the migration of neutral or protonic molecules of MA in the frame of MAPbI₃ crystal with fraction of MAI inducement.

I conclude on this investigation, first of all, authors should be defined the crystal and chemical properties on MAPbI₃ with fraction of MAI. Second, the enhancement of solar cell efficiency is corresponding to primarily iodide supplementary by adding MAI and protonic MA can be suspending the inorganic frame. These two combinations are able to make improving solar cell performances. The residual Ac anion can probably locates to defect site of iodide in MAPbI₃. The migration factor of MA can be attributed to neutral MA and is not protonic MA. The defect behaviors as beta, delta and gamma in DLTS can be remarkably related with the phases of MA (forming neutral to protonic) with increasing iodide concentration. When authors can explain well in this correlation, many parts of interpretations will change.

Report Reviewer #1:

-As shown in Figure S4 the capacitance, and therefore also permittivity is strongly frequency and temperature dependent due to the ion movement. Equations (4) and (5) contain the permittivity, it is not clear which permittivity values are used for the slow and fast(er) ions and how the temperature dependence of the permittivity is taken into account in the activation energy for the ion diffusion (Eq. (4)).

We thank reviewer #1 for the response and for the assessment that our study is interesting. The referee is right to point out that the information about the exact value used for further calculation is missing. The permittivity in this study is determined from the negative voltage component of the CV measurement for high AC frequency of $f = 80$ kHz ($\omega \approx 500$ kHz), from which it can be assumed that there is complete depletion. In this case, the capacitance for reverse bias agrees with the geometrical capacitance and allows a calculation of the permittivity.

To emphasize that, we added the following sentences to Sec. IIA:

The relative permittivity can be obtained at reverse bias in the CV measurements where complete depletion can be assumed. In this case the capacitance in this region corresponds to the geometrical capacitance C_{geo} , which can be used to calculate the relative permittivity as follows:

$$C_{\text{geo}} = \frac{\epsilon_0 \epsilon_R}{d} \quad (2)$$

The obtained values are shown in Fig. S3b.

Accordingly, we modified Fig. S3, which shows additionally the used permittivity obtained by CV measurements:

Fig. S3. a) Capacitance-voltage measurement on the perovskite solar cells with variation of precursor stoichiometry at room temperature with an ac frequency of 80 kHz. The data were evaluated according to the Mott–Schottky approach by plotting $1/C^2$ and fitting the linear part at around 0.8 V which correspond to the depletion capacitance. b) Relative Permittivity can be calculated from geometrical capacitance C_{geo} obtained from negative voltage part of a) and film thickness d by AFM measurements using Eqn. (2).

It is noteworthy that the capacitance at reverse bias (CV measurement at high frequency, Fig. 3a) is a better approach than trying to determine the geometrical capacitance from the high frequency plateau of the C_f measurements at 0 V (Fig. S4). In the case of reverse bias and high frequencies, the CV measurements are usually almost independent of the frequency, since all charge carriers are driven to the interfaces of the transport layers leaving behind a completely depleted space charge. In this case, the capacitance only depends on the width of the space charge zone extending over the entire perovskite layer. Additionally, at very high frequencies, the charge carriers cannot follow the AC voltage and therefore do not contribute to the capacitance. Under the conditions of reverse bias and high frequencies, the geometrical capacitance can be determined adequately, and can be used to calculate the permittivity (Fig. S3 b, top).

According to Fig. S4, while the frequency dependence of the CV measurement can be neglected for high frequencies in the range of $\omega \approx 500$ kHz, the temperature dependence has an impact on the results obtained by CV measurements. This impact, while present, is very small. To illustrate the magnitude of change, please consider the following calculation using Eqn. 2 (with $d = 300$ nm):

A temperature dependent change of the capacitance of $\Delta C = 4 \cdot 10^{-9}$ F/cm² as, for instance, for the high frequency component of the stoichiometric sample (see Fig. S4) would lead to a change in relative permittivity of $\Delta \epsilon_R = 1.3$, $\frac{\Delta \epsilon_R}{\epsilon_R} = 6\%$. In short, the temperature

dependence of the permittivity is small and has only a marginal effect on calculation of the migration rates.

We added the following sentences to the manuscript:

In the case of high frequencies, the temperature dependence of the capacitance is small and can be neglected.

-The amount of ions calculated (EQ. (5), Figure 3) are only in the 10^{14} - 10^{15} cm⁻³ concentration range. This is considerably lower than the amount of charge carriers generated in a solar cell. As a result, effects of ions on band bending and built-in voltage would be overruled by movement of the charge carriers which would compensate these effects. What is missing as a validation of the obtained numbers are numerical calculations of the band bending and electric field profiles.

We thank the reviewer for raising this point. He/She is right to note that it was reported that the defect density have to exceed charge carrier densities of roughly 10^{16} cm⁻³ according to [10.1016/j.joule.2019.10.003], so that ions can have an impact on the electronic landscape. Different models are presented in that paper, whereby either one or two mobile ionic species can be present. However, the authors only assume one mobile ionic species for their calculation of the band bending. Several studies (10.1038/ncomms8497, 10.1039/C9MH00445A) including ours have shown experimentally that there are several mobile ionic species. In our manuscript, we justify (at the end of the results section) that the fast and slow response in DLTS/IS originate from different mobile species, respectively. The difference in the magnitude of the defect density results, therefore, from the choice of the model. In order to emphasize this, we estimate the defect densities required to shift the built-in potential as shown Fig. 1. The model with only one mobile species according to Almora et al. (10.1021/acs.jpcclett.5b00480) gives

$$N_{\text{ion}} = \frac{\Delta V_{\text{bi}}^2 C_{\text{C}}^2}{\epsilon_{\text{R}} \epsilon_0 k_{\text{B}} T},$$

where ΔC corresponds to the capacitance step of the IS spectra and ΔV_{bi} refers to the potential drop by ion accumulation in accordance with our calculation of Eqn. (8). No double layer capacitance is assumed here, i.e. no serial connection of the capacitive contributions of different ionic species. We performed this estimate as an example for the lowest (2.96) and highest (3.06) stoichiometry, assuming that only cations are mobile. For the stoichiometry of 2.96 with $\Delta V_{\text{bi}} = 90$ mV, the ionic defect density has to be in the range of $N_{\text{ion}} = 5 \cdot 10^{14}$ cm⁻³, which is more than one order of magnitude higher than the ion density determined by IS with the two-ion model used in our manuscript for this sample. In the case of the stoichiometry of 3.06 with $\Delta V_{\text{bi}} = 260$ mV the expected defect density would be $N_{\text{ion}} = 2 \cdot 10^{18}$ cm⁻³, which is two orders of magnitude higher compared with the model with two mobile ionic species. In summary, we state that the ion densities in the one-ion model, which is often used in simulations for band bending, require higher ion density in order to have an influence on the band bending. In the two-ion model, however, a smaller ion density is already sufficient, as the serial connection of two double-layer capacitances is taken into account. This in itself is an important information which should be made available to the community.

Furthermore, it is noteworthy that the dark charge carrier density in MAPbI₃ perovskite solar cells is a sensitive parameter and was found— depending on the exact stoichiometry, the manufacturing methods and fabrication parameters—to be in the range from 10¹⁰ cm⁻³ to 10¹⁸ cm⁻³ (10.1038/s41560-018-0324-8, 10.1063/1.4899051, 10.1016/j.nanoms.2019.10.004). We could not determine the actual dark charge carrier density for our samples, since the determined doping density is most likely overestimated by the ion concentration, as explained in detail in the manuscript in Sec. IIA. Since the measurements were carried out in the dark and therefore no photogenerated charge carriers are expected, we assume, however, that the actual charge carrier density is on the same scale as the ion density, and band bending therefore occurs. The shift of the built-in voltage as an experimental observable of the CV measurement confirms this assumption, as our correction of this shift using the measured ion densities, is quite successful (see Fig. 1). Furthermore, in the paper by Fassel et al. (10.1039/c8ee01136b) a band bending of around 0.4 eV at the interfaces was already shown by UPS measurements for a series of samples made identically to the ones in our study. This serves as a validation of our results, since we showed by CV measurements a comparable shift in total of around 350 mV similar to the finding in that previous work. Finally, it is worth noting that we do not see any evidence for a complete screening of the internal electric field. Instead, an ion-induced band bending near the interfaces is likely due to the above-mentioned correction of the built-in potential giving realistic values compatible with literature.

We changed the manuscript accordingly:

We note that we use a model assuming two mobile ionic species to correct the shift of the built-in potential in contrast to several studies where only one mobile species is assumed. [10.1021/acs.jpcclett.5b00480, 10.1016/j.joule.2019.10.003] Accordingly, the magnitude of the defect density necessary to cause band bending is lower compared to these studies (for more information see SI).

In the supplemental information we added a section to highlight the new insight:

X Understanding the impact of small ion densities on the electronic landscape

As reported in several studies,[10.1021/acs.jpcclett.5b00480, 10.1016/j.joule.2019.10.003] ion densities in the range of the charge carrier density have an impact on the electronic landscape by causing interfacial band bending and partly screening the internal electric field. To calculate the voltage drop caused by ion accumulation (i.e. the ion density N_{ion}), it is very important whether only one species, e.g. cations, or both species (anions and cations) are considered to be mobile. Accordingly, one can differentiate between a one-ion model and a two-ion model, respectively. Eqn. (9) represents the potential drop by the two-ion model, whereas for the one-ion model an approach without series connection of the double layer capacitance has been found[10.1021/acs.jpcclett.5b00480]:

$$\Delta V_{\text{bi}} = \frac{\sqrt{N_{\text{ion}} \epsilon_{\text{R}} \epsilon_0 k_{\text{B}} T}}{\Delta C} . \quad (\text{S1})$$

ΔC corresponds to the capacitance step of the IS spectra and ΔV_{bi} refers to the potential drop by ion accumulation in accordance with our calculation of Eqn. (8). We have demonstrated this calculation for the sample with the largest potential drop with a stoichiometry ratio of 3.06

according to our calculation with Eqn. (9) (see Fig. (1)) and, for comparison, with the one-ion model:

Tab S11: Calculation of the potential drop by using the two-ion model according to Eqn. (9). b) Calculation of the ion density using the one-ion model and the same potential drop as in a). c) Comparison with Almora et al.[10.1021/acs.jpcllett.5b00480].

	a) Two-ion model (Eqn. (9))	b) One-ion model (Eqn. (S1))	c) One-ion model with estimation by Almora et al. [10.1021/acs.jpcllett.5b00480]
N_{ionC}	$1.9 \cdot 10^{16} \text{ cm}^{-3}$	$2.7 \cdot 10^{18} \text{ cm}^{-3}$	$2.4 \cdot 10^{17} \text{ cm}^{-3}$
N_{ionA}	$1.3 \cdot 10^{14} \text{ cm}^{-3}$		
ΔC_C	$4.1 \cdot 10^{-7} \text{ F/cm}^2$	$4.1 \cdot 10^{-7} \text{ F/cm}^2$	$2 \cdot 10^{-6} \text{ F/cm}^2$
ΔC_A	$3.4 \cdot 10^{-8} \text{ F/cm}^2$		
ϵ_R	19	19	32.5
ΔV_{bi}	326 mV	326 mV	27 mV

All calculations were carried out for room temperature. We calculated the necessary ion density for the one-ion model in column b) to yield the same potential drop ΔV_{bi} as determined in the two-ion model a) (Fig. 1), by keeping the experimental observable ΔC and ϵ_R constant. As shown in column a) and b) of Tab. S11, the ion densities can be significantly different to impact the electronic landscape expressed by the same voltage drop, depending on which model was applied for the respective perovskite. The difference originates from the fact that in the case of the one-ion model no double layer capacitance is assumed, i.e. no serial connection of the capacitive contributions of different ionic species. In column c) we compared our calculation with the estimation of the voltage drop according to Almora et al.[10.1021/acs.jpcllett.5b00480]. It shows the influence of space charge capacitance in relation to ion density on the calculation of the potential drop and confirms our findings.

Report Reviewer #2:

Careful measurements are done on devices, rather than clean materials, to deduce information on the active, perovskite material, which is assumed to be clean in and out because of its mode of preparation, which, even though it probably affects the surfaces and interfaces rather than the bulk, does appear to do so controllably!

We thank reviewer #2 for considering the manuscript and providing insights from his/her perspective on our results. We agree with the reviewer that the ion behavior between application-related devices and „clean“ materials might be different. We studied devices on

purpose, as we are convinced that the measurement of ionic defects *in devices* is more relevant for the development of the solar cell technology. The discussion between interface and bulk defects indeed seems to be more complex for perovskite solar cells than for other thin film technologies. In our study, we refer to the model of the accumulation of ions at the interfaces to the transport layers. This model has been used and proven more frequently in the literature and describes the relationship of ion migration with regard to bulk and interfaces well (see 10.1039/C9MH00445A, 10.1038/ncomms13831, 10.1021/acs.jpcllett.5b02229). With respect to our study, it can be shown that the model used is clearly well suited to describe ion migration and its influence on the electronic landscape, as the trend of the built-in potential caused by ion accumulation can be corrected well (see Fig. 1).

DLTS and halide lead perovskites are both hard to control, as materials, and as a method, because of the need to prepare the systems in an initial state that is well-controlled and the need for contacts. The ref. cited suggests such a method and notes its results on the FA Pb perovskite, and its problem with the MA one; the onus of showing that here the result that presumably is obtained, a p-n junction, remains stable during all of the measurements, is on the authors.

Our study is based on the results of the publication by Fassel et al.(10.1039/c8ee01136b, 10.1039/c8tc05998e) We use a batch of samples that were produced under exactly the same conditions. The study by Fassel et al. shows that the preparation of samples for this stoichiometric variation is reproducible. Besides this, we measured two different batches with the reported stoichiometric variation, and both show consistent results and prove repeatability. In addition, JV measurements were performed both before and after the characterization by defect spectroscopy to verify that the samples did not degrade in the meantime.

To emphasize this, we added these sentences to the method section:

JV measurements were performed both before and after characterization by defect spectroscopy to verify that the samples did not degrade during the experiment.

All measurements were performed on two different batches for proving repeatability.

The main problem with the report is that it fails to distinguish between clear experimental observables and inferred phenomena, something that, admittedly, is hard and is a general issue in much of scientific research, but seems here to be quite significant. A simple example is the caption of Fig. S1, which contains 2 sentences, one about observables and the other a conclusion that gives, though, the impression of being an observable.

We agree with the reviewer that it is important to distinguish between observables and their interpretation. We have checked our manuscript carefully and changed the following sentences, which are stated below:

- We deleted the sentences that hysteresis indicates the existence of mobile ions in the caption from Fig. S1. (SI)

- In results part A: "According to literature,[56-58] the increase in N_{eff} suggests an overall higher defect density for overstoichiometric samples since ions introduce additional charges and affect the net doping concentration."

- In results part B: "We conclude that by increasing the sample stoichiometry, ion migration of defect γ is suppressed, while the defect β becomes more mobile."

- In the caption of Fig. 2: "Areas are shaded differently in order to highlight where each defect is observable."

The samples are likely polycrystalline films with a certain grain size distribution, possibly with a respectable area to volume ratio. This can make the interfaces a significant part of the film and, while this is certainly known, the conclusions that are drawn give the impression as referring to the defect physics of the bulk material. If that is not the case, that is not clear.

According to the results of Fassel et al. (10.1039/c8ee01136b), the microstructure of the films (processed like our samples) with varying stoichiometry is very similar, with the grain size changing from the lowest stoichiometry to the highest by only 7 %. Therefore, we do not expect that this small change in the grain size could have a high impact on the defect landscape such that it could change the defect concentration and diffusion coefficients over several orders of magnitude (see Fig. 3).

We changed the discussion section accordingly:

We note that the observed trends in activation energy, diffusion coefficient and defect density cannot be explained by morphology changes, since measurements on an identically made set of samples show only a slightly change of the grain (see Fassel et al. [10.1039/c8ee01136b]) which cannot be the origin of the shift in the defect parameters over several orders of magnitude.

The Meyer-Neldel rule part shows the data that suggest this effect. However, the authors come up against what has been the main problem with this rule, a clear physical explanation that is testable and gives new insight. In the discussion (there is a problem re. ref. 85, which is for inorganic materials) disorder is suggested and multi-phonon excitation. Possible, but maybe it is possible to first test it against a more intuitive possibility that can be found in papers from 1968, 1970 and 1986, by B Rosenberg, Kemeny & Rosenberg and JC Dyle, respectively? After that fails, then the above-noted two options can be further considered and content be provided for the statement that they "offer interesting insights into the physical mechanisms of ion migration in halide perovskites".

Our report of the Meyer-Neldel rule for ionic defects is a novel finding for metal halide perovskites, and we believe that the corresponding data is quite convincing and helpful to understand the spread of defect activation energies found in literature. Furthermore, for the first time it allows a classification/assignment of the defects found in literature. We agree with the reviewer concerning the need for a clear physical explanation, but point out that a final test

cannot always be provided in the publication observing the effect. The reviewer states that ref. 85 and the model proposed within cannot be used, since he/she assumes that perovskite behaves as an organic semiconductor. However, looking at the similar observations of hybrid and inorganic perovskite thin films, we point out that the metal halide perovskites behave rather as inorganic than as organic semiconductors. In contrast, the studies mentioned by the author relate very strongly to organic materials, where the applicability to perovskites might be limited. For instance, Rosenberg et al. 1968 (10.1063/1.1670724) considered oxidized cholesterol, retinal, and nucleic acids, and no clear explanation is given for the origin of the Meyer-Neldel rule. As several studies show (10.1021/acs.jpcclett.7b03414, 10.1038/nature16977, 10.1088/0034-4885/69/4/R04), multi-phonon excitation is a more likely origin of the Meyer-Neldel rule for the studied perovskite materials than tunneling through intermolecular potential barriers as described by Kemeny et al. 1970 (10.1063/1.1670724). A glass transition, as stated as the cause of the Meyer-Neldel rule in the study by Dyre 1986 (10.1088/0022-3719/19/28/016), has not yet been shown in perovskite materials. In the preparation of our manuscript, we discussed several options for the origin of the Meyer-Neldel rule in perovskites. We are convinced that considering multi-phonon excitation and a distribution of ion migration rates to be two credible options for explaining the origin of the Meyer-Neldel rule in perovskites materials.

None of this, though, need in the end, prevent publishing this work, after shrinking, including making parts less verbose, and clearly separating experimental observables from the rest.

We thank the reviewer for the evaluation that our manuscript should be published. We considered it carefully in terms of clarity and conciseness, and revised it accordingly.

Remains that it is hard to understand why this should be in Nat. Comm. (also after re-reading the purpose of the journal). Why not submit to an Applied Physics journal, or if the word “applied” is to be shunned, to PRMaterials, for example?

We are convinced that our observations on ionic defects and their properties in metal halide perovskites are unique and striking results. Our observation about the clear trend of the ionic defect parameters for very small variations of the stoichiometric enables a direct correlation to the processing conditions in metal halide perovskites. Analyzing the impact of mobile ions on the electronic landscape is very important to understand phenomena such as hysteresis and degradation observed in hybrid perovskites. The assignment of ionic defects proposed by us, applying the Meyer-Neldel rule we observed for the first time in perovskite solar cells, allows to put the findings in literature into a more general context. Finally, with this revision we were also able to carve out that even smaller ion densities can have a strong impact on the energetic landscape (as observed by the apparent built-in potential) when more than one ion species is mobile. Our work is well within the scope of the Nature Communications and would be of great interest for its readership.

Report Reviewer #3:

This investigation is trying to figure out the behaving defect formation and the ionic

migration in MAPbI₃ with the controlling minor concentrations of MAI molecule inducement by EIS and DLTS analysis. The resulting values are explained in Meyer-Neldel rule which is very traditional theory for microscopic explanation of compensation on activation energy increase. I am sure this investigation that has high novelty of understanding defect formation and migration behavior on MAPbI₃. This article is major revision as mentioned below list.

We thank the referee for the positive assessment of our work and the recognition that it is of high novelty. Having taken into consideration the comments provided by the referee, we made improvements in a number of areas, which are detailed below.

1. Authors do not show the crystal properties by SEM and XRD. In particular, the SEM surface images are remarkably an important for obtained DLTS results. Because of, the trap concentrations at interface between electrode and perovskite layer is also one of key factor to enhance of deep trap level. (Science 20 Mar 2020: Vol. 367, Issue 6484, pp. 1352-1358). The perovskite film morphologies with fractions of MAI with PbAc₂ should be providing them in SI. When the film morphologies are looks very similar or same, we can begin to discuss on DLTS and EIS results. I can expect the significant morphology differences based on PCE distributions with fraction of MAI. Please should be defined film morphologies.

We thank referee for raising the important issue of microstructure. In our previous studies exploring for the first time the effect of stoichiometric variations on the properties of the films (see Fassel et al. (10.1039/c8ee01136b, 10.1039/c8tc05998e), we already investigated in detail the microstructure of the films by SEM and XRD. We found that variation in stoichiometry does not change the microstructure and all films exhibit very similar grain size and crystal structure. In this study, we focus on investigating the ion migration in devices which were made identically to the samples investigated by us before (i.e. they were processed with the same recipe under exactly the same conditions). Accordingly, the findings of those two studies also apply to our work. Detailed SEM, XRD and film morphology analysis can be found for instance in Fig. 2, 4 in Ref. 10.1039/c8ee01136b.

To highlight this better, we added the following sentence to the Sec. II:

For more information about detailed SEM, XRD and film morphology analysis see Ref. [10.1039/c8ee01136b].

2. Authors have used fraction of MAI 2.96 to 3.06, which is changed very tiny concentration of MAI for formation of MAPbI₃. The controlling MAI concentration as addressed quantities should be leading to change MA and I defect sites in our expectation. However, the formation of MAPbI₃ using precursor is not able to define as our expectation, for example, the various fraction of MAI with a concentration of PbAc₂ compounds in polar solvent is on stoichiometric to non-stoichiometric status. 1 : 3 of PbAc₂ : MAI is stoichiometric precursor but others are not. The stoichiometric of elemental coordination is not related with using fraction between MAI and PbAc₂ compounds. The most important point of perovskite material before this investigation

have to be defined chemical analysis first. When you can define the differences of chemical quantification on organic, Pb and I, DLTS results can explain. Please shows quantified elements on the consisting chemicals on MAPbI₃ using XPS or another possible chemical analysis.

Chemical analysis using XPS measurements on identically samples to determine the chemical composition in dependence of stoichiometry were done by Fassel et al. (10.1039/c8ee01136b)

We changed the manuscript accordingly:

Chemical analysis for verification of the composition change with stoichiometry were performed by Fassel et al. [10.1039/c8ee01136b] on a series of identical samples.

3. In previous investigation (J. Am. Chem. Soc. 2020, 142, 13, 6251–6260), the very small concentration of tri-iodide addition as less than 50 mM is appeared to form dense MAPbI₃ film. It is also obtained to enhance PCE and Voc as addressed in author's investigation. I can presume that MAI addition into MAPbI₃ is leading to increase iodide site in the perovskite crystal comparing to defect passivation by MA site.

According to the studies of Fassel et al.(10.1039/c8ee01136b, 10.1039/c8tc05998e) and the herein made XPS/UPS measurements on a series of identically made samples, we expect a gradual increase of I⁻ and MA⁺ interstitials and a decrease of I⁻ and MA⁺ vacancies with increasing precursor stoichiometry.

4. If authors will have obtained full chemical analysis, you will be understanding the residual Ac molecules are remained in MAPbI₃ film. This molecule can be playing role as reduction of anion site defect. But we do not know the remained concentrations of Ac molecules with fraction of MAI. If you can find out this point, I promise this investigation will be excellent for more wide audience in this research area.

We thank the reviewer for raising this point, and thank him/her for evaluating our manuscript as „excellent“ in this research area. It has been well established that methylammonium acetate, MA(OAc) is completely removed during the thermal annealing process of the perovskite layer due to its low thermal stability and the inability of acetate to be incorporated into the perovskite lattice due to its small ionic radius (10.1039/c8ee01136b, 10.1038/ncomms7142, 10.1039/C6TA09554B). This has been shown experimentally by different methods. Correspondingly, the resulting films are therefore pure MAPbI₃ and Ac molecules do not remain in the film and thus do not impact our findings in any way.

We modified the manuscript accordingly:

We note that no Ac molecules remain in the perovskite films as those are removed during the annealing resulting in the formation of high quality polycrystalline MAPbI₃ films [10.1039/c8ee01136b, 10.1038/ncomms7142, 10.1039/C6TA09554B].

5. One more point of view, the localized MA in MAPbI₃ has two kinds of phases as protonic and neutral molecules. (Appl. Phys. Lett. 108, 073901 (2016)), (J. Am. Chem. Soc. 2017, 139, 46, 16462–16465) The located those MA molecules are being on different

defect formations. In particular, two MA molecule can be acting dynamics with different temperatures of environments such as 200K to 350K. The dynamics of MA from order to disorder phases are contributed to change inorganic crystal frames from disorder to order. Namely, DLTS results should be considered to interpret this point of view the migration of neutral or protonic molecules of MA in the frame of MAPbI₃ crystal with fraction of MAI inducement.

We thank the reviewer for this insight. We point out, though, that we are experimentally not able to detect neutral species. With our methods (IS and DLTS), the internal field is manipulated either by an ac alternating voltage and/or a voltage pulse, so that the electrically charged ions migrate in accordance with the change in the internal field. Neutral ions cannot be influenced by the electric field changes, so that an additional consideration of these protonic ions would be pure speculation on our part. Furthermore, as stated in the manuscript, we want to clarify that our assignment to the type of ion is based on a comparison to the literature. Our DLTS method can only differentiate between the charge type of the ions (+/-) and not between the ion types. Additionally, with the measured migration rates, which differed over several orders of magnitude, and the charge type of the defects recovered, one would not expect that the origin of these defects could be traced back to protonic MAI or MAI ions alone.

We added the following sentences to the manuscript to emphasize this:

We notice furthermore that only electrically charged species can be observed. We therefore exclude neutral protonic species of MAI as reported in literature (10.1021/jacs.7b09319) to be the origin of the observed mobile ions.

REVIEWER COMMENTS

Reviewer #3 (Remarks to the Author):

To authors

According to your revises, I thanks to you for all your answers from my questions.

I hope that your research as the investigation of defect in perovskite materials with the advanced theory will be well progressing yourself.

This investigation will also be able to extend for broad audiences in the area of perovskite materials.

Best regards,

Reviewer #4 (Remarks to the Author):

Reichert et al. investigate ionic defects in MAPbI₃ perovskites by impedance spectroscopy (IS) and deep-level transient spectroscopy (DLTS). By adjusting the precursor stoichiometry of the perovskites, they change the quantity of defects and thus of mobile ions within the films. They show that even small ion densities are sufficient to considerably affect the electronic landscape and quantify the effect of mobile ion density on the screening of the built-in potential. They find that the ionic defects follow the Meyer-Neldel rule. They give two possible explanations for this behaviour – multi-phonon excitation and a distribution of ion migration rates. Impressively, their results can explain the wide range of ion migration rates found in the literature.

The findings are novel and contribute to a fundamental understanding of ionic effects in metal halide perovskites. The authors have convincingly addressed all concerns of the reviewers. In particular, I appreciate the added discussion about the effect of the mobile ion density on the potential drop. This is a much-discussed topic and this work will help advance knowledge about the effects of mobile ions on the electronic landscape in metal halide perovskites.

I think that the manuscript is well suited for Nature Communications. I suggest publishing this work in Nature Communications after some additional small modifications, as stated below.

DLTS by Lang et al. and admittance spectroscopy by Walter et al. are two methods originally developed to quantify electronic defects in semiconductors, and not ionic defects. The authors state that "the defect density from IS measurements cannot be determined by using the approach by Walter et al. for semiconductor defects." I fully agree with the authors that this method is not applicable because ionic effects are clearly visible in the impedance spectra. However, in the presence of electronic defects, I would expect effects from electronic defects to overlap with effects from ionic ones. Similarly, in DLTS, electronic defects should also be visible in case they are present. How can the authors be sure that they measure ionic defects and not electronic ones?

While the results of DLTS and IS are in very good agreement, the methods probe slightly different regions inside the perovskite. In DLTS, ion migration within the bulk of the perovskite is probed. In IS at 0 V DC bias, ions are located at the interfaces to screen the built-in potential. Therefore, when applying a small voltage perturbation, I would expect that ion migration is predominantly probed near the interfaces. Could the difference in activation energy between IS and DLTS perhaps be explained by the difference in ion migration near the interfaces and inside the bulk? In addition, I wonder why one of the defects is only visible in DLTS and not in IS?

To probe ionic defects with DLTS, the authors apply a filling bias for 100 ms. Assuming that the ions are located at the interfaces before the filling bias and that the migration of ions into the perovskite bulk is governed by diffusion rather than drift, the diffusion coefficient would have to be in the order of $10^{-9} \text{ cm}^2\text{s}^{-1}$ in order to reach a new steady state. While this is true for two of the measured defects, one has a much slower diffusion coefficient in the order of $10^{-11} \text{ cm}^2\text{s}^{-1}$ at 300 K. Although I do not expect this to have a strong influence on the obtained diffusion coefficient and activation energy, this could have an influence on the measured mobile ion density. The authors should discuss this point in more detail.

The authors should mention the scan rate and the temperature at which the current-voltage measurements were performed.

Moritz H. Futscher

Reviewer #5 (Remarks to the Author):

The manuscript by Reichenhall et al. studies the defect landscape in halide perovskite solar cells via impedance and deep level transient spectroscopy. The investigated samples were device-like samples where fractional alternations of MAI were undertaken when processing the samples. That those small fractional alterations may play quite a significant role has been demonstrated before and form the base of the herein presented work, offering a systematic way to study defects. The others find the prevalence of 3 defect species with different migration rates. A rather phenomenological description with the Meyer-Neldel rule has been presented.

The manuscript has already undergone a thorough revision from 3 previous reviewer reports with seemingly all different backgrounds which significantly contributed to the improvement of the manuscript. The authors have thoroughly responded to the comments of the reviewers.

The manuscript fits well into the scope of Nat. Comm. which is well-read by the perovskite solar cell community and the presented work and issues will attract a broad readership.

I recommend the publication of this manuscript.

Second Report of Referee #4

We thank the referee for the assessment that our study is novel and contribute to the fundamental understanding of ionic defects in metal halide perovskites.

(1) DLTS by Lang et al. and admittance spectroscopy by Walter et al. are two methods originally developed to quantify electronic defects in semiconductors, and not ionic defects. The authors state that "the defect density from IS measurements cannot be determined by using the approach by Walter et al. for semiconductor defects." I fully agree with the authors that this method is not applicable because ionic effects are clearly visible in the impedance spectra. However, in the presence of electronic defects, I would expect effects from electronic defects to overlap with effects from ionic ones. Similarly, in DLTS, electronic defects should also be visible in case they are present. How can the authors be sure that they measure ionic defects and not electronic ones?

We agree with the referee that this is an important issue. It is certainly possible for ionic and electronic signals to overlap (i.e., to show similar emission/migration rates at a given temperature). However, the defects we have determined are all of ionic nature, according to reverse DLTS measurements we performed in our previous work (10.1103/PhysRevApplied.13.034018). To address this issue in the manuscript, we added the following sentences: "Since we performed reverse-DLTS measurements in our recent work to distinguish between electronic and ionic defects,[10.1103/PhysRevApplied.13.034018] we attribute all observed defects to mobile ions." This does not mean that electronic defects are not present, but that either their density is too low to be observed by our methods, considering the dominant ionic signals, or that the electronic defects are energetically shallow, as has been shown in several theoretical studies (10.1063/1.4864778, 10.1021/acs.chemmater.5b01909).

(2) While the results of DLTS and IS are in very good agreement, the methods probe slightly different regions inside the perovskite. In DLTS, ion migration within the bulk of the perovskite is probed. In IS at 0 V DC bias, ions are located at the interfaces to screen the built-in potential. Therefore, when applying a small voltage perturbation, I would expect that

ion migration is predominantly probed near the interfaces. Could the difference in activation energy between IS and DLTS perhaps be explained by the difference in ion migration near the interfaces and inside the bulk? In addition, I wonder why one of the defects is only visible in DLTS and not in IS?

We thank the referee for his insight into this important point. The scenario described by the referee is a plausible explanation for the difference between IS and DLTS and can complement our explanation in the manuscript that different experimental methods can probe different parts of the same ionic defect distribution (according to our recent work 10.1103/PhysRevApplied.13.034018). We have added this explanation to the manuscript: **“Furthermore, the difference between IS and DLTS can be caused by the fact that the mobile ions with IS are detected when they reside near the interfaces at 0 V DC bias, whereas with DLTS, the mobile ions are probed while they move from the bulk to the interface.”**

By having a closer look into the IS data of the sample with e.g. 3.04 stoichiometry (as shown in the following picture for temperatures around 300 K), the defect δ is poorly observable for a very narrow temperature range and with a low peak height. A fit over these two or three migration rates would lead to a too large error in activation energy and diffusion coefficient to be comparable with the DLTS results where defect δ can be observed clearly separated from the other defects. For lower stoichiometric ratios, the defect δ is even less visible as shown in Fig. 2. For this reason we decided that defect δ cannot be evaluated with IS and we used only the DLTS data. To state this more clearly, we modified the manuscript accordingly: **“The remaining defect with comparably high migration rates was labeled δ . This defect cannot be evaluated with IS as it is only observable as a shoulder, not a peak, and only for higher stoichiometric ratios in a very narrow temperature range.”**

(3) To probe ionic defects with DLTS, the authors apply a filling bias for 100 ms. Assuming that the ions are located at the interfaces before the filling bias and that the migration of ions into the perovskite bulk is governed by diffusion rather than drift, the diffusion coefficient would have to be in the order of $10^{-9} \text{ cm}^2\text{s}^{-1}$ in order to reach a new steady state. While this is true for two of the measured defects, one has a much slower diffusion coefficient in the order of $10^{-11} \text{ cm}^2\text{s}^{-1}$ at 300 K. Although I do not expect this to have a strong influence

on the obtained diffusion coefficient and activation energy, this could have an influence on the measured mobile ion density. The authors should discuss this point in more detail.

We agree that due to the chosen pulse length and considering the small diffusion constant of defect γ , not all ions of this species may reach equilibrium. In this case the measured ion density of this species would only represent a lower limit. We added the following sentences to the manuscript: “Due to the fact that the diffusion coefficient of γ is comparably small, it cannot be ensured that this ionic species reaches equilibrium within the duration of the filling pulse. As a result, the determined ion density for γ represents only a lower limit.” A study with pulse width scans on perovskite solar cells/single crystals is planned and will hopefully provide new insights to estimate the influence of the pulse width on the determined defect density.

(4) The authors should mention the scan rate and the temperature at which the current-voltage measurements were performed.

We added these important information to the method section with the following sentence: “All measurements were performed at room temperature (300 K) with a scan rate of 0.25 V/s.”